# Adaptive-saturated RNN:
# Remember more with less instability

**Khoi Minh Nguyen-Duy[1], Quang Pham[2], Thanh Binh Nguyen[1,3]**
[1]Department of Mathematics and Computer Science, VNUHCM - US, Vietnam
[2]Institute for Infocomm Research (I[2]R), A*STAR, Singapore
[3]AISIA Research Lab
Corresponding author: ngtbinh@hcmus.edu.vn

## ABSTRACT

Orthogonal parameterization is a compelling solution to the vanishing gradient problem (VGP) in recurrent neural networks (RNNs). With orthogonal parameters and non-saturated activation functions, gradients in such models are constrained to unit norms. On the other hand, although the traditional vanilla RNNs are seen to have higher memory capacity, they suffer from the VGP and perform badly in many applications. This work proposes Adaptive-Saturated RNNs (asRNN), a variant that dynamically adjusts its saturation level between the two mentioned approaches. Consequently, asRNN enjoys both the capacity of a vanilla RNN and the training stability of orthogonal RNNs. Our experiments show encouraging results of asRNN on challenging sequence learning benchmarks compared to several strong competitors. The research code is accessible at https://github.com/ndminhkhoi46/asRNN/.

## 1 MOTIVATION

Training vanilla RNNs (with tanh activation) is notoriously challenging due to the VGP, where the gradients' magnitudes become *exponentially* smaller (Collins et al., 2017). Thus, extensive efforts have been devoted to develop more stable, effective models such as orthogonal RNNs (Lezcano-Casado & Martínez-Rubio, 2019), and LSTM (Hochreiter & Schmidhuber, 1997). Empirically, orthogonal RNNs show more competitive performances compared to LSTM and vanilla RNN on long-sequence tasks. However, due to non-saturated activations and unitary constrain, their memory capacity is also more limited compared to vanilla RNN (Collins et al., 2017).

This study aims to realize the memory capacity potential of vanilla RNNs by endowing them with the capability to address the VGP of orthogonal RNNs. As a result, we propose the *adaptive-saturated RNN* (asRNN), a vanilla RNN variant that dynamically adjusts the activation's saturation level. Particularly, we observe that, let $f(x; a) = \frac{\tanh(ax)}{a}$, then:

$$\lim_{a \to 0} f(x; a) = x, \quad \text{and} \quad \lim_{a \to 1} f(x; a) = \tanh(x). \tag{1}$$

Thus, by generalizing and using $f(x; a)$ as an activation function, we can adjust $a$ and update the parameters freely of a vanilla RNN to achieve high memory capacity without being affected by the VGP. In the following, we will formally introduce asRNN and outline a key result of a condition for avoiding the VGP in asRNN.

## 2 METHODOLOGY

**Formulation** Based on the observation Eq. 1, we formally define the hidden cell of asRNN as:

$$h_t = \boldsymbol{W}_f^{-1}\tanh(\boldsymbol{W}_f(\boldsymbol{W}_{xh}\boldsymbol{x}_t + \boldsymbol{W}_{hh}\boldsymbol{h}_{t-1} + \boldsymbol{b})), \tag{2}$$

where $\mathcal{W} = \{\boldsymbol{W}_f, \boldsymbol{W}_{xh}, \boldsymbol{W}_{hh}, \boldsymbol{b}\}$ is the set of trainable parameters, $\boldsymbol{W}_f$ introduces an end-to-end composite layer to control the saturation level of asRNN. To ensure the non-singularity of $\boldsymbol{W}_f$, we parameterize $\boldsymbol{W}_f = \boldsymbol{U}_f\boldsymbol{D}_f$, where $\boldsymbol{U}_f$ is orthogonal, $\boldsymbol{D}_f$ is positively diagonal. Remarkably, we observe that: (i) by fixing $\boldsymbol{W}_f$ to be identity, we recover a vanilla RNN; and (ii) let $\boldsymbol{W}_{hh}$ be orthogonal, fix $\boldsymbol{U}_f$ to be the identity, and let $\boldsymbol{D}_f \to \mathbb{0}$, we recover an orthogonal RNN. From such construction, asRNN not only dynamically adjusts the saturation level but also controls the singular values of the temporal Jacobian (Thm. 2.1), which in turn alleviates the VGP.

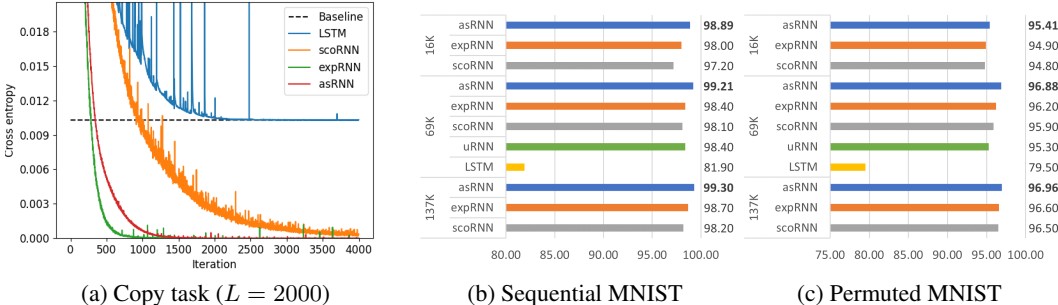

(a) Copy task ($L = 2000$)     (b) Sequential MNIST     (c) Permuted MNIST

Figure 1: Training loss for the Copy task and test accuracies for the sequential and permuted MNIST.

**Key theoretical result** Let $w \in \mathcal{W}$, we define the gradient and Jacobian as (Pascanu et al., 2012): $\frac{\partial \mathcal{L}}{\partial w} = \sum_{1 \le t_1 \le T} \frac{\partial \mathcal{L}}{\partial h_T} J(T, t_1) \frac{\partial h_{t_1}}{\partial w}$, where $J(t_2, t_1) = \prod_{t_1 < t \le t_2} J(t)$ and $J(t) = W_f^{-1} \text{diag}[1 - (W_f h_t)^2] \bar{W}_f W_{hh}$. The vanishing gradient problem in RNNs is credited to the existence of a temporal Jacobian $J(t_2, t_1) \approx 0$ that bottlenecks the backpropagated signal. Under mild assumptions (Appendix A.2), we show a condition for asRNN where all singular values of temporal Jacobian matrices are lower bounded by 1. Thanks to this, the VGP is alleviated on vanilla RNN.

**Theorem 2.1.** *Let $G$ and $H$ be respectively the $d_h$-th degree generalized permutation group and its signed permutation subgroup. Under the assumptions in Appendix A.2, if $||D_f||_2 \le$*

$$\frac{arctanh(\sqrt{1 - ||W_{hh}^{-1}||_2})}{(||W_{xh}||_2 C_x + ||b||_\infty) \sum_{i=0}^{t-1} (||W_{hh}||_{\max} + 1)^i} \text{ and } \min_{E \in G} ||W_{hh} - E||_2 \le \frac{\sigma_{\min}(D_f)}{||D_f||_2}, \text{ then}$$

$$\forall t_2 > t_1 \in \mathbb{N}^*, \exists \epsilon \ge 0 : \min_{E \in H} ||U_f - E||_2 \le \epsilon \to \sigma_{\min}(J(t_2, t_1)) \ge 1.$$

Importantly, while Zhao et al. (2020) questioned and showed a scenario where vanilla RNN and LSTM have a short memory, a problem associated with the VGP. In contrast, our Thm. 2.1 suggests the existence of another in which vanilla RNN resists VGP and possesses long memory.

## 3 EXPERIMENT

To validate the model's memory capacity, we consider the Copy task(Hochreiter & Schmidhuber, 1997), sequential MNIST(LeCun et al., 1998), and permuted MNIST(Goodfellow et al., 2014). Next, we use the Penn Treebank character-level prediction (PTB-c)(Marcus et al., 1993) task to explore the model's expressivity (Kerg et al., 2019; Bojanowski et al., 2016). We benchmark asRNN against the strong orthogonal RNNs with long memories and high trainability such as expRNN(Lezcano-Casado & Martínez-Rubio, 2019), scoRNN(Helfrich et al., 2018), uRNN(Arjovsky et al., 2016), and the popular LSTM(Hochreiter & Schmidhuber, 1997) with high expressivity from the gated mecha-

| Model | $T = 150$ | $T = 300$ |
|---|---|---|
| LSTM | **$1.41 \pm 0.005$** | **$1.43 \pm 0.004$** |
| asRNN | $1.46 \pm 0.006$ | $1.49 \pm 0.005$ |
| nnRNN | $1.47 \pm 0.003$ | $1.49 \pm 0.002$ |
| expRNN | $1.49 \pm 0.008$ | $1.52 \pm 0.001$ |
| EURNN | $1.61 \pm 0.001$ | $1.62 \pm 0.001$ |
| RNN-orth | $1.62 \pm 0.004$ | $1.66 \pm 0.006$ |
| RNN | $2.89 \pm 0.002$ | $2.90 \pm 0.002$ |

Table 1: Test BPC on PTB-c at different BPTT lengths (T). Best results are in bold.

nism. We follow the setting described in Appendix A.4 and report the results in Fig. 1 and Tab. 1. On long sequence learning tasks (Fig. 1), our asRNN shows excellent performance by converging stably on the Copy task, and achieves better generalization on the sequential and permuted MNIST tasks. On the PTB task, asRNN achieved encouraging results by outperforming all orthogonal RNN baselines, only second to LSTM. Overall, our empirical results show that asRNN possesses both high memory capacity and expressivity compared to other non-gated RNNs and can alleviate the EVGP, which corroborates our motivation in Sec. 1.

## 4 CONCLUSION

We have investigated the potential and limitations of vanilla RNNs in learning long-sequence tasks. Then, we proposed asRNN, a novel vanilla RNN variant that enjoys strong resistance to the VGP and can possess long memory. Our experiment results show asRNN enjoys encouraging performances on tasks that demand memory span, memory capacity, or expressivity.

ACKNOWLEDGEMENT

This research is supported by the National Research Foundation, Singapore under its AI Singapore Programme (AISG Award No: AISG2-RP-2021-027).

URM STATEMENT

Author Khoi Minh Nguyen-Duy meets the URM criteria of the ICLR 2023 Tiny Papers Track.

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

## A  APPENDIX

### A.1  RELATED WORK

**Long short-term memory.** A dominant approach in preventing the exploding and vanishing gradient problem in RNN is with gated mechanisms, such as LSTM (Hochreiter & Schmidhuber, 1997). This model is easily trained and gives good performance on a variety of sequence learning tasks. Despite the fact that LSTM has a lower memory capacity compared to vanilla RNN (Collins et al., 2017), and that it performs badly on long-memory tasks (Helfrich et al., 2018), we include this model for its high effectiveness in language modeling, where the decisive factor is expressivity such as mechanism or hidden size (Bojanowski et al., 2016).

**Orthogonal RNN.** This class of models addressed the VGP with non-saturated activation functions (e.g. modReLU) and a unitary hidden weight matrix. To constrain the unitarity during gradient descent, scoRNN followed a scaled Cayley transformation, while uRNN and EURNN used pre-selected unitary transformation compositions (Helfrich et al., 2018; Arjovsky et al., 2016; Jing et al., 2017). To compensate for the expressivity loss, expRNN proposed to use the computationally cheap exponential map stemming from Lie group theory (Lezcano-Casado & Martínez-Rubio, 2019). As a result, the gradients grow only *linearly* in sequence length. There is also an initialization scheme proposed for this class to directly solve many long sequence learning tasks (Henaff et al., 2016).

**Non-normal RNN.** Despite having high trainability, being unitary limits the expressivity of Orthogonal RNNs compared to their free parameters counterparts. Non-normal RNN, which leverages Schur decomposition to acquire non-unit eigenvalues, is proposed to overcome this problem while maintaining the trainability of Orthogonal RNNs (Kerg et al., 2019). Consequently, nnRNN outperforms Orthogonal RNNs on both long sequence learning tasks and language modeling tasks.

### A.2  ASSUMPTIONS

**Assumption A.1.** *(scale-independent loss)* $\exists \epsilon > 0, \forall \delta > 0 : ||\boldsymbol{y} - \hat{\boldsymbol{y}}||_2 \geq \delta \wedge \mathcal{L}(\boldsymbol{y}, \hat{\boldsymbol{y}}) < \epsilon$.

Assumption A.1 prevents the convergence from closing the gap between $||\hat{\boldsymbol{y}}||_2$ and $||\boldsymbol{y}||_2$ by re-scaling the output layer with a factor of $||\boldsymbol{D}_f||_2$, which potentially causes the VGP. As a remark, it is possible to reduce this effect on scale-dependent loss with $\mathcal{L}(\delta^{-1}\boldsymbol{y}, \hat{\boldsymbol{y}})$, for some small hyperparameter $\delta > 0$. We leave this for consecutive works.

**Assumption A.2.** *(bounded input distribution)* $\exists C_x > 0, \forall t : ||\boldsymbol{x}_t||_\infty \leq C_x$.

In practice, it is common to normalize input with $C_x = 1$ before training neural network-based architectures. Assumption A.2 prevents the absurdity of an infinitely large input saturating asRNN and causing the VGP.

**Assumption A.3.** *(no nilpotent singular value)* $\sigma_{\min}(\boldsymbol{W}_{hh}) \geq 1$.

Assumption A.3 is unnecessary in practice even when $\boldsymbol{W}_{hh}$ is orthogonally constrained thanks to the amplification of gradients using $\boldsymbol{W}_f^{-1}$ (see Appendix A.4.1).

### A.3 PROOF OF THEOREM 2.1

**Lemma A.1.** *David Kewei Lin (2019) For a diagonal matrix $\boldsymbol{D}$ and a conformable matrix $\boldsymbol{A}$, we have $||\boldsymbol{D}\boldsymbol{A}||_2 \leq ||\boldsymbol{D}||_{\max}||\boldsymbol{A}||_2$.*

**Lemma A.2.** *Let $G$ and $H$ be respectively the $d_h$-th degree generalized permutation group and its signed permutation subgroup. Under the assumptions in Appendix A.2, if $||\boldsymbol{D}_f||_2 \leq$*

$$\frac{arctanh\left(\sqrt{1 - ||\boldsymbol{W}_{hh}^{-1}||_2}\right)}{(||\boldsymbol{W}_{xh}||_2 C_x + ||\boldsymbol{b}||_\infty)\sum_{i=0}^{t-1}(||\boldsymbol{W}_{hh}||_{\max} + 1)^i} \text{ and } \min_{\boldsymbol{E}\in G}||\boldsymbol{W}_{hh} - \boldsymbol{E}||_2 \leq \frac{\sigma_{\min}(\boldsymbol{D}_f)}{||\boldsymbol{D}_f||_2}, \text{ then}$$

$$\forall t \in \mathbb{N}^*, \exists \epsilon(t) \geq 0 : \min_{\boldsymbol{E}\in H}||\boldsymbol{U}_f - \boldsymbol{E}||_2 \leq \epsilon(t) \rightarrow \sigma_{\min}(\boldsymbol{J}(t)) \geq 1.$$

*Proof.* Let $\boldsymbol{E}_{hh} = \arg\min_{\boldsymbol{E}\in G}||\boldsymbol{W}_{hh} - \boldsymbol{E}||_2$ and $\boldsymbol{R}_{hh} = \boldsymbol{W}_{hh} - \boldsymbol{E}_{hh}$. First we show that $||\boldsymbol{W}_f\boldsymbol{h}_t||_\infty$ is bounded:

$$||\boldsymbol{W}_f\boldsymbol{h}_t||_\infty = ||\tanh(\boldsymbol{W}_f(\boldsymbol{W}_{xh}\boldsymbol{x}_t + \boldsymbol{W}_{hh}\boldsymbol{h}_{t-1} + \boldsymbol{b}))||_\infty \tag{3}$$

$$= \tanh(||\boldsymbol{W}_f(\boldsymbol{W}_{xh}\boldsymbol{x}_t + \boldsymbol{W}_{hh}\boldsymbol{h}_{t-1} + \boldsymbol{b})||_\infty) \tag{4}$$

$$\leq \tanh(||\boldsymbol{W}_f\boldsymbol{W}_{xh}\boldsymbol{x}_t||_\infty + ||\boldsymbol{W}_f\boldsymbol{W}_{hh}\boldsymbol{h}_{t-1}||_\infty + ||\boldsymbol{W}_f\boldsymbol{b}||_\infty) \tag{5}$$

$$\leq \tanh\left(||\boldsymbol{D}_f||_2(||\boldsymbol{W}_{xh}||_2 C_x + ||\boldsymbol{b}||_\infty) + ||\boldsymbol{D}_f\boldsymbol{W}_{hh}\boldsymbol{D}_f^{-1}||_2||\boldsymbol{W}_f\boldsymbol{h}_{t-1}||_\infty\right) \tag{6}$$

$$\leq \tanh\left(||\boldsymbol{D}_f||_2(||\boldsymbol{W}_{xh}||_2 C_x + ||\boldsymbol{b}||_\infty)\sum_{i=0}^{t-1}||\boldsymbol{D}_f(\boldsymbol{E}_{hh} + \boldsymbol{R}_{hh})\boldsymbol{D}_f^{-1}||_2^i\right) \tag{7}$$

$$\leq \tanh\left(||\boldsymbol{D}_f||_2(||\boldsymbol{W}_{xh}||_2 C_x + ||\boldsymbol{b}||_\infty)\sum_{i=0}^{t-1}\left(||\boldsymbol{W}_{hh}||_{\max} + \frac{||\boldsymbol{R}_{hh}||_2||\boldsymbol{D}_f||_2}{\sigma_{\min}(\boldsymbol{D}_f)}\right)^i\right) \tag{8}$$

$$\leq \tanh\left(||\boldsymbol{D}_f||_2(||\boldsymbol{W}_{xh}||_2 C_x + ||\boldsymbol{b}||_\infty)\sum_{i=0}^{t-1}(||\boldsymbol{W}_{hh}||_{\max} + 1)^i\right) \tag{9}$$

$$\leq 1 - \frac{1}{\sigma_{\min}(\boldsymbol{W}_{hh})} \tag{10}$$

Let $\boldsymbol{E}_f = \arg\min_{\boldsymbol{E}\in H}||\boldsymbol{U}_f - \boldsymbol{E}||_2, \boldsymbol{R}_f = \boldsymbol{U}_f - \boldsymbol{E}_f, \boldsymbol{D}_t = \text{diag}[1 - (\boldsymbol{W}_f\boldsymbol{h}_t)^2]$. Then

$$\sigma_{min}(\boldsymbol{J}(t)) = \sigma_{\min}\left(\boldsymbol{D}_f^{-1}\boldsymbol{U}_f^\mathsf{T}\text{diag}\boldsymbol{D}_t\boldsymbol{U}_f\boldsymbol{D}_f\boldsymbol{W}_{hh}\right) \tag{11}$$

$$\geq \sigma_{\min}\left(\boldsymbol{D}_f^{-1}\left(\boldsymbol{R}_f^\mathsf{T} + \boldsymbol{E}_f^\mathsf{T}\right)\boldsymbol{D}_t(\boldsymbol{R}_f + \boldsymbol{E}_f)\boldsymbol{D}_f\right)\sigma_{\min}(\boldsymbol{W}_{hh}) \tag{12}$$

$$\geq \left(\sigma_{\min}\left(\boldsymbol{D}_f^{-1}\boldsymbol{E}_f^\mathsf{T}\boldsymbol{D}_t\boldsymbol{E}_f\boldsymbol{D}_f\right) - ||\boldsymbol{D}_f^{-1}\left(\boldsymbol{E}_f^\mathsf{T}\boldsymbol{D}_t\boldsymbol{R}_f + \boldsymbol{R}_f^\mathsf{T}\boldsymbol{D}_t\boldsymbol{E}_f + \right.\right.$$
$$\left.\left.\boldsymbol{R}_f^\mathsf{T}\boldsymbol{D}_t\boldsymbol{R}_f\right)\boldsymbol{D}_f||_2\right)\sigma_{\min}(\boldsymbol{W}_{hh}) \tag{13}$$

$$\geq \left(1 - \max_i(\boldsymbol{W}_f\boldsymbol{h}_t)_i^2 - \frac{||\boldsymbol{D}_f||_2^2}{\sigma_{\min}(\boldsymbol{D}_f)}\left(2||\boldsymbol{R}_f||_2 + ||\boldsymbol{R}_f||_2^2\right)\right)\sigma_{\min}(\boldsymbol{W}_{hh}) \tag{14}$$

$$\geq \left(\frac{1}{\sigma_{\min}(\boldsymbol{W}_{hh})} - \frac{||\boldsymbol{D}_f||_2^2}{\sigma_{\min}(\boldsymbol{D}_f)}||\boldsymbol{R}_f||_2\right)\sigma_{\min}(\boldsymbol{W}_{hh}) \tag{15}$$

The rest of the proof follows directly from the previous inequality. □

**Theorem 2.1.** *Let $G$ and $H$ be respectively the $d_h$-th degree generalized permutation group and its signed permutation subgroup. Under the assumptions in Appendix A.2, if $||\boldsymbol{D}_f||_2 \leq \frac{arctanh(\sqrt{1-||\boldsymbol{W}_{hh}^{-1}||_2})}{(||\boldsymbol{W}_{xh}||_2 C_x + ||\boldsymbol{b}||_\infty) \sum_{i=0}^{t-1}(||\boldsymbol{W}_{hh}||_{\max}+1)^i}$ and $\min_{\boldsymbol{E}\in G} ||\boldsymbol{W}_{hh} - \boldsymbol{E}||_2 \leq \frac{\sigma_{\min}(\boldsymbol{D}_f)}{||\boldsymbol{D}_f||_2}$, then*

$$\forall t_2 > t_1 \in \mathbb{N}^*, \exists \epsilon \geq 0 : \min_{\boldsymbol{E}\in H} ||\boldsymbol{U}_f - \boldsymbol{E}||_2 \leq \epsilon \rightarrow \sigma_{\min}(\boldsymbol{J}(t_2, t_1)) \geq 1.$$

*Proof.* From Lemma A.1 and Lemma A.2, if $\epsilon = \min_{t_1 \leq t \leq t_2} \epsilon(t)$, then $||\boldsymbol{U}_f - \boldsymbol{E}||_2 \leq \epsilon \rightarrow \sigma_{\min}(\boldsymbol{J}(t_2, t_1)) \geq 1$ □

## A.4 EXPERIMENTAL SUPPLEMENTARY

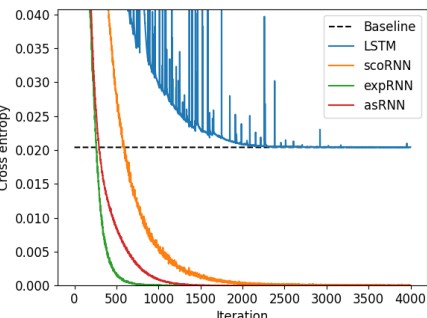

Figure 2: Copy task ($L = 1000$).

| Model | T=150 | T=300 |
|---|---|---|
| LSTM | **69.81 ± 0.001** | **69.60 ± 0.0003** |
| asRNN | 68.93 ± 0.001 | 68.59 ± 0.001 |
| nnRNN | 68.78 ± 0.001 | 68.52 ± 0.0004 |
| expRNN | 68.07 ± 0.15 | 67.58 ± 0.04 |
| RNN-orth | 66.29 ± 0.07 | 65.53 ± 0.09 |
| EURNN | 65.68 ± 0.002 | 65.55 ± 0.002 |
| RNN | 40.01 ± 0.026 | 39.97 ± 0.025 |

Table 2: Test accuracy on PTB-c at different BPTT lengths (T). Best results are in bold.

| Model | $L = 1000$ | $L = 2000$ |
|---|---|---|
| LSTM | 0.40 ± 0.02 | 0.78 ± 0.03 |
| scoRNN | 0.70 ± 0.02 | 1.35 ± 0.04 |
| expRNN | 0.62 ± 0.03 | 1.20 ± 0.04 |
| asRNN | 0.70 ± 0.02 | 1.33 ± 0.04 |

| #params | sMNIST | pMNIST |
|---|---|---|
| 16K | 98.89 | 95.41 |
| 69K | 99.21 | 96.88 |
| 137K | 99.30 | 96.96 |

Table 3: Iteration train time in seconds at Copy Memory

Table 4: Iteration train time in seconds at Copy Memory

### A.4.1 COMMON SETTING

The batch size across all experiments was 128. asRNN is optimized using smoothing constant $\alpha = 0.99$ on MNISTs tasks, and $\alpha = 0.9$ otherwise. We report the best results on a similar setup to ours in Jing et al. (2017), Arjovsky et al. (2016), Helfrich et al. (2018), Lezcano-Casado & Martínez-Rubio (2019), and Kerg et al. (2019). For other experiments, we replicate the same setup of Lezcano-Casado & Martínez-Rubio (2019) (copy memory and pixelated MNIST), and of Kerg et al. (2019) (Penn Treebank character-level prediction).

Since no advantage was observed when $\boldsymbol{W}_{hh}$ is updated freely according to Assumption A.3 during preliminary empirical testing, this layer is kept strictly orthogonal for the maximum memory-per-parameter capacity. As a result, both $\boldsymbol{U}_f$ and $\boldsymbol{W}_{hh}$ are parameterized as described in Lezcano-Casado & Martínez-Rubio (2019). The diagonal matrix is parameterized as $\boldsymbol{D}_f = \text{diag}(\boldsymbol{d}_f)$, where $\boldsymbol{d}_{f_i} = |s_i| + \epsilon, \boldsymbol{s} \in \mathbb{R}^{d_h}, \epsilon > 0$.

According to Theorem 2.1, we initialize $\boldsymbol{U}_f, \boldsymbol{W}_{hh}$ close to a permutation matrix, such as the identity or those in Henaff et al. (2016) and Helfrich et al. (2018). When $a, b,$ and $\epsilon$ are close to zero, asRNN resembles a linear RNN. For this reason, we find the hyperparameter setting of expRNN to be compatible with asRNN. Thus, $\ln(\boldsymbol{W}_{hh})$ uses Henaff initialization (Henaff et al., 2016) for copy memory tasks and Cayley initialization (Helfrich et al., 2018) for other tasks. We also initialize

$\boldsymbol{W}_{xh}$ as random semi-orthogonal and $\boldsymbol{b}$ as zero for a greater bound of $\boldsymbol{D}_f$. Ultimately, we initialize $\boldsymbol{s} \sim \mathcal{U}(x; a, b)$, where the optimal settings of $a, b, \epsilon$ are task-dependent.

Not only we can avoid the VGP by setting up $a, b, \epsilon$ close to zero, but we can also avoid the exploding gradient problem (EGP) by setting $a, b, \epsilon$ far from zero (See PTB-c in Table 5). As mitigating the EGP is not our main focus, we use gradient clipping with norm 10 on pixelated MNIST and copy memory tasks, then we select the best hyperparameters for asRNN.

### A.4.2   COPY MEMORY TASK

Copy memory is a synthetic many-to-many classification task first introduced in Hochreiter & Schmidhuber (1997) to test the ability to recall information bits after a very long delay. This task is designed to be extremely difficult for the vanilla RNN due to the VGP (Henaff et al., 2016).

Input samples contain the alphabets 2-9, and the blank and start letters 0, 1, are one-hot encoded into vectors of dim 10. The recalling sequence $\mathcal{A}$ contains the first $K$ letters of the input and are uniformly sampled from the alphabet. It is followed by $L$ blank letters, 1 start letter, and another $K-1$ blank letters. An output sample contains $K+L+1$ blanks letter and the recalling sequence $\mathcal{A}$. The baseline model output $K+L+1$ blanks letter and another $K-1$ letters sampled uniformly of the alphabet, which has the loss of $\frac{K\ln 8}{L+2K}$.

On this task, we benchmark asRNN against the architectures: LSTM, scoRNN, and expRNN. The recalling length $K$ is 10 for both of the experiments in Figure 2 and Figure 1a). The seed on this task is set at 5544 and the number of parameters is fixed at 22K. All experiments are optimized with smoothing constant $\alpha = 0.9$.

We include in Table 3 the average iteration training time in seconds of Copy Memory experiments. This result shows asRNN achieved similar computational complexity when compared to scoRNN, and was marginally worse than expRNN. It is also worth noting that LSTM achieved low running time due to its native implementation in Pytorch, while other methods were implemented from scratch. Overall, asRNN achieved similar training time compared to other orthogonal RNNs, while offering promising performance improvements, especially when solving problems requires long-term memories.

### A.4.3   PIXELATED MNIST

Pixelated MNIST is a classification task for hand-written numbers (LeCun et al., 1998). It benchmarks RNN sequence learning under a continuous stream of input.

On the sequential MNIST task (sMNIST, unpermuted pixelated MNIST), each $28 \times 28$ image of a decimal digit is pixelated into a 784 input sequence, then is fed into the RNN to produce an embed vector used for classification. On the permuted MNIST task (pMNIST, permuted pixelated MNIST), a permutation is applied to all samples before they are pixelated.

On this task, we benchmark asRNN against the following architecture: LSTM, scoRNN, expRNN, restricted-capacity unitary RNN (uRNN)(Arjovsky et al., 2016). The seed on this task is set at 5544.

Total memory capacity is one among the properties that is limited by hidden sizes. For example, orthogonal RNNs might possess infinitely long memory despite having small hidden size, but are limited in the total memory capacity.

We excerpt Fig. 1 for a memory capacity sensitivity analysis over the number of parameters for benchmarks. The summarized results for asRNN in Table 4 shows that under the same sequence length, as the total number of parameters increases, so is the total memory capacity.

### A.4.4   PENN TREEBANK CHARACTER-LEVEL PREDICTION (PTB-C) TASK

Penn Treebank is a corpus that has an alphabet size of 50, including the end of sentence symbol, and is first introduced in (Marcus et al., 1993). We choose this many-to-many character-level modeling task because rather than bottlenecking RNNs' memory capacity, it tests their expressiveness, which includes structural design and the hidden size Bojanowski et al. (2016). Gated mechanisms such as LSTM is known to be effective for PTB-c.

On this task, we benchmark asRNN against the following architecture: LSTM, expRNN, nnRNN (Kerg et al., 2019), EURNN (Jing et al., 2017), modReLU RNN with Glorot initialization (RNN), modReLU RNN with orthogonal initialization (RNN-orth). For the Penn Treebank character-level prediction, each experiment is carried out for 5 repeated trials. On this task, the number of parameters for each model is fixed at 1.32M.

## B  FUTURE WORK

In the future, we plan to extend asRNN to scale-dependent loss (AssumptionA.1). We also aim to reduce the computational cost using non-exponential sigmoids such as $\frac{x}{(1+|x|^k)^{k-1}}$.

As our work focuses on addressing the long memory and memory capacity problems of vanilla RNNs, we were unable to directly compare with relatively recent architecture such as UniCORNN for one key reason: they have multiple layers and more complex architectures such as stacked layers, gates, etc. In contrast, all methods we considered are related to the vanilla RNN with a single layer to demonstrate the main purpose of this study. In future works, we will consider improving asRNN for a fair comparison with more recent architectures.

| Task | #epoch/iter | $d_h$ | lr | lr $\boldsymbol{W}_{hh}$ | $a$ | $b$ | $\epsilon$ |
|---|---|---|---|---|---|---|---|
| Copy memory | 4000 | 138 | $2 \cdot 10^{-4}$ | $10^{-4}$ | 0 | 0 | $2 \cdot 10^{-5}$ |
| sMNIST | 70 | 122 | $7 \cdot 10^{-4}$ | $7 \cdot 10^{-5}$ | $2 \cdot 10^{-2}$ | $2 \cdot 10^{-2}$ | $10^{-2}$ |
| sMNIST | 70 | 257 | $5 \cdot 10^{-4}$ | $5 \cdot 10^{-5}$ | $2 \cdot 10^{-2}$ | $2 \cdot 10^{-2}$ | $10^{-2}$ |
| sMNIST | 70 | 364 | $3 \cdot 10^{-4}$ | $3 \cdot 10^{-5}$ | $2 \cdot 10^{-2}$ | $2 \cdot 10^{-2}$ | $10^{-2}$ |
| pMNIST | 70 | 122 | $10^{-3}$ | $10^{-4}$ | $2 \cdot 10^{-2}$ | $2 \cdot 10^{-2}$ | $10^{-2}$ |
| pMNIST | 70 | 257 | $7 \cdot 10^{-4}$ | $7 \cdot 10^{-5}$ | $2 \cdot 10^{-2}$ | $2 \cdot 10^{-2}$ | $10^{-2}$ |
| pMNIST | 70 | 364 | $7 \cdot 10^{-4}$ | $7 \cdot 10^{-5}$ | $2 \cdot 10^{-2}$ | $2 \cdot 10^{-2}$ | $10^{-2}$ |
| PTB-c (T=150) | 100 | 1024 | $10^{-3}$ | $10^{-3}$ | $8 \cdot 10^{-1}$ | 3 | 0 |
| PTB-c (T=300) | 100 | 1024 | $10^{-3}$ | $10^{-3}$ | $8 \cdot 10^{-2}$ | 3 | 0 |

Table 5: Hyperparameters for asRNN.

