# OpenReview forum: "Adaptive-saturated RNN: Remember more with less instability"
_ICLR.cc/2023/TinyPapers — Submitted to Tiny Papers @ ICLR 2023_

### Official Review · Reviewer_u6ar · 2023-03-29

**Confidence:** 3

**Summary Of Contributions:**

This paper proposes the Adaptive-Saturated RNN (asRNN) model, which can dynamically adjust the activation function saturation, so as to achieve high memory capacity and not be affected by the VGP.

**Rating:**

Clear, Correct, and Reproducible (CCR): a submission which meets the reviewing criteria

**Strengths And Weaknesses:**

# Strengths
1. This paper proposes a new RNN variant, asRNN, which does not suffer from the vanishing gradient problem while having high memory capacity and expressive power.
2. The method in this paper is simple and elegant, and the method is proved in principle.
3. Extensive experiments demonstrate that asRNN is as effective and reliable as baseline methods in terms of long-term memory.

# Weaknesses
1. First of all, I believe that this article has not been compared with RNN variants that are relatively recent in time, such as UniCORNN. The baseline methods are relatively outdated and cannot fully explain the advantages of asRNN.
2. The paper does not have a comparative experiment on the time cost of asRNN and baseline methods in training and reasoning, which is a pity.
3. It is well known that the long-term memory of RNN is limited by its hidden size. Can the authors consider adding parameter sensitivity experiments on the hidden size?
4. The structure of the paper is incomplete, and the main body even lacks the Section on *Conclusion*.

**Suggested Changes:**

In general, I think this paper is a solid work, but it needs to consider the four points mentioned in the *Weakness* to further optimize the paper. Simultaneously, in view of  [the submission requirements of Tiny Paper](https://iclr.cc/Conferences/2023/CallForTinyPapers): the paper should not exceed two pages and the practice of moving the majority of the content into appendices is discouraged, I prefer that this paper can be optimized and published as a long paper instead of a Tiny paper .

---

> ### Author Response · Authors · 2023-04-22
> **Response to the Official Review of Paper27 by Reviewer u6ar**
>
> We thank the reviewer for their detailed feedback. We are encouraged that they find asRNN to be a solid work, and a simple and elegant method.
>
> Q1: "The paper does not have a comparative experiment on the time cost of asRNN and baseline methods in training and reasoning, which is a pity."
>
> A1: Thank you for your constructive feedback. Unfortunately, as we only reuse the best reported results on MNIST and PTB-c, we don’t have access to their time cost in training and reasoning. However, we have included in the revision the **average iteration training time in seconds** of Copy Memory experiments. For your convenience, we also provide the results below.
>
> |  Model |    L=1000   |    L=2000   |
> |:------:|:-----------:|:-----------:|
> | LSTM   | 0.40 ± 0.02 | 0.78 ± 0.03 |
> | scoRNN | 0.70 ± 0.02 | 1.35 ± 0.04 |
> | expRNN | 0.62 ± 0.03 | 1.20 ± 0.04 |
> | asRNN  | 0.70 ± 0.02 | 1.33 ± 0.04 |
>
> This result shows that asRNN achieved quite similar computational complexity compared to scoRNN, and was marginally worse than expRNN. It is also worth noting that LSTM achieved low running time due to its native implementation in Pytorch, while other methods were implemented from scratch. Overall, our asRNN achieved similar training time compared to other orthogonal RNNs, while offering promising performance improvements, especially when solving problems requires long-term memories.
>
> Q2: “It is well known that the long-term memory of RNN is limited by its hidden size. Can the authors consider adding parameter sensitivity experiments on the hidden size?”
>
> A2: Thank you for your insightful suggestion on adding parameter sensitivity experiments. Rather than memory span, we believe that total memory capacity is one of the properties limited by hidden sizes. For example, orthogonal RNNs possess infinitely long memory despite having small hidden size, but are limited in total memory capacity.
>
> We have done sensitivity analysis over the number of parameters for memory capacity benchmarks (MNIST tasks) in Fig 1.b and 1.c. Here is the summarized results of asRNN:
>
> | #params | sMNIST | pMNIST |
> |:-------:|:------:|:------:|
> | 16K     | 98.89  | 95.41  |
> | 69K     | 99.21  | 96.88  |
> | 137K    | 99.30  | 96.96  |
>
> This experiment shows that under the same sequence length, as the total number of parameters increases, so is the total memory capacity.
>
> Q3: “The structure of the paper is incomplete, and the main body even lacks the Section on Conclusion.”
>
> A3: We apologize for this mistake. In the revision, we have included the Conclusion section into the main text.
>
> Q4: “First of all, I believe that this article has not been compared with RNN variants that are relatively recent in time, such as UniCORNN. The baseline methods are relatively outdated and cannot fully explain the advantages of asRNN.”.
>
> A4: Thank you for the constructive suggestion. As our work focuses on addressing the long memory and memory capacity problems of vanilla RNNs, we were unable to directly compare with relatively recent architecture such as UniCORNN for one key reason: they have multiple layers and more complex architectures such as stacked layers, gates, etc. In contrast, all methods we considered are related to the vanilla RNN with a single layer to demonstrate the main purpose of this study. Due to the current limitations in time and space, we will leave the work of improving asRNN for a fair comparison with more recent architectures in the future.
>
> Q5: “I prefer that this paper can be optimized and published as a long paper instead of a Tiny paper.”
>
> A5: Thank you for the suggestion. We will definitely further improve our work to publish it as a long paper.

---

### Official Review · Reviewer_DVB4 · 2023-04-01

**Confidence:** 3

**Summary Of Contributions:**

This work proposes Adaptive-Saturated RNNs (asRNN), which combines the capacity of vanilla RNNs with the training stability of orthogonal RNNs. The asRNN dynamically adjusts the saturation level between these two types of RNNs, allowing it to avoid the vanishing gradient problem and perform well on challenging sequence learning benchmarks.

**Rating:**

High Potential (HP): a submission which meets the reviewing criteria and has potential to make an impact on the field

**Strengths And Weaknesses:**

### Strengths:
1. The paper is well-motivated, clear and the proposed method is supported by theoretical proofs.

### Weaknesses:
1. My only concern is about the applicability of this work in applied settings. Can the authors explain more on areas where asRNN might offer an advantage over LSTMs specifically?

**Suggested Changes:**

1. How many experimental trials were used for reporting the mean and standard error in Tables 1?
2. "Unless specified otherwise, asRNN is optimizes using RMSprop with α = 0.99" - Does this imply that the other methods were optimized using some other optimizer setting?

---

> ### Author Response · Authors · 2023-04-19
> **Response to Official Review of Paper27 by Reviewer DVB4**
>
> Q1:“Can the authors explain more on areas where asRNN might offer an advantage over LSTMs specifically?”
>
> A1: Thank you for the question. One direct application where orthogonal RNNs have been observed to outperform LSTMs are speech processing [M]. In our work, asRNNs have higher memory capacity than orthogonal RNNs and longer memory than LSTMs. This might reduce the need for downsampling signals and increasing the window size in tasks concerning high-frequency information such as speech recognition. Thereby potentially increase performance by preventing information loss in such task.
>
> [M] Mario Lezcano-Casado and David Martınez-Rubio. Cheap orthogonal constraints in neural networks:
> A simple parametrization of the orthogonal and unitary group. In Kamalika Chaudhuri and Ruslan
> Salakhutdinov (eds.), Proceedings of the 36th International Conference on Machine Learning,
> volume 97 of Proceedings of Machine Learning Research, pp. 3794–3803. PMLR, 09–15 Jun
> 2019.
>
> Q2: “How many experimental trials were used for reporting the mean and standard error in Tables 1?”
>
> A2: The results in Table 1 are reported over five trials.
>
>
> Q3: “"Unless specified otherwise, asRNN is optimizes using RMSprop with α = 0.99" - Does this imply that the other methods were optimized using some other optimizer setting?”
>
> A3: As for the optimization method used for the other methods. asRNN is optimized using smoothing constant alpha=0.99 on MNISTs tasks, and alpha=0.9 in other experiments.
>
> We have revised the optimizer’s information in the writing. Thank you again for your valuable time and effort in reviewing our paper.

---

### Author Response · Authors · 2023-04-19
**General Response to all AC and Reviewers**

We appreciate the AC and Reviewers for the valuable feedback. We are delighted that the Reviewers recognized the contributions of our work. They agreed that our method is simple, elegant, and supported by theoretical results.

We have taken the Reviewers’ feedback to further improve our work. Specifically, the revision of our work is concluded as follows.
- We revised the paper so that it fits within the 2 pages limit.
- We revised the paper so that the Conclusion is now in the main text.
- We clarified that "asRNN is optimized using smoothing constant alpha=0.99 on MNISTs tasks, and alpha=0.9 otherwise" in the paper.
- We included parameter analysis for memory capacity over hidden size.
- We included train time for Copy Memory and its discussion.
- We included the plan for improving asRNN and benchmarking it with more recent models in Future Directions.
- We fixed various typos and improved the overall presentation.

We now address each Reviewer’s concern individually.

---

### Author Response · Authors · 2023-05-30
**Opt-in for Archival**

We would like to opt-in for archival. Thank you!

---

### Meta-Review · Area_Chair_gP9X · 2023-04-06

**Recommendation:** Invite to present
**Confidence:** 5

**Metareview:**

 This work combines the capacity of vanilla RNNs with the training stability of orthogonal RNNs and develops an asRNN. Generally, this work is clear and solid. however, the presentation is not good. Also, extensive experiments are needed to be done.

**Summary:**

This is an interesting paper to improve training stability of orthogonal RNNs.

**Comments And Feedback To The Authors:**

See the meta review.

**Reason For Not Giving A Higher Recommendation:**

N?A

**Reason For Not Giving A Lower Recommendation:**

This work is novel and solid.

---

### Decision · Program_Chairs · 2023-04-10

**Decision:**

Invite to present

**Comment:**

Slightly over two pages. Please correct in camera-ready.